# Short-Term Microgravity Influences Cell Adhesion in Human Breast Cancer Cells

**DOI:** 10.3390/ijms20225730

**Published:** 2019-11-15

**Authors:** Mohamed Zakaria Nassef, Sascha Kopp, Daniela Melnik, Thomas J. Corydon, Jayashree Sahana, Marcus Krüger, Markus Wehland, Thomas J. Bauer, Christian Liemersdorf, Ruth Hemmersbach, Manfred Infanger, Daniela Grimm

**Affiliations:** 1Clinic for Plastic, Aesthetic and Hand Surgery, Otto von Guericke University Magdeburg, 39120 Magdeburg, Germany; mohamed.nassef@med.ovgu.de (M.Z.N.); daniela.melnik@med.ovgu.de (D.M.); marcus.krueger@med.ovgu.de (M.K.); markus.wehland@med.ovgu.de (M.W.); thomas.bauer@med.ovgu.de (T.J.B.); manfred.infanger@med.ovgu.de (M.I.); 2Department of Biomedicine, Aarhus University, 8000 Aarhus C, Denmark; corydon@biomed.au.dk (T.J.C.);; 3Department of Ophthalmology, Aarhus University Hospital, 8200 Aarhus N, Denmark; 4Institute of Aerospace Medicine, Department of Gravitational Biology, German Aerospace Center, 51147 Cologne, Germany; christian.liemersdorf@dlr.de (C.L.); ruth.hemmersbach@dlr.de (R.H.); 5Gravitational Biology and Translational Regenerative Medicine, Faculty of Medicine and Mechanical Engineering, Otto von Guericke University, 39120 Magdeburg, Germany

**Keywords:** breast cancer cells, microgravity, hypergravity, cell adhesion, apoptosis, NF-κB

## Abstract

With the commercialization of spaceflight and the exploration of space, it is important to understand the changes occurring in human cells exposed to real microgravity (r-µ*g*) conditions. We examined the influence of r-µ*g*, simulated microgravity (s-µ*g*, incubator random positioning machine (iRPM)), hypergravity (hyper-*g*), and vibration (VIB) on triple-negative breast cancer (TNBC) cells (MDA-MB-231 cell line) with the aim to study early changes in the gene expression of factors associated with cell adhesion, apoptosis, nuclear factor “kappa-light-chain-enhancer” of activated B-cells (NF-κB) and mitogen-activated protein kinase (MAPK) signaling. We had the opportunity to attend a parabolic flight (PF) mission and to study changes in RNA transcription in the MDA-MB cells exposed to PF maneuvers (29th Deutsches Zentrum für Luft- und Raumfahrt (DLR) PF campaign). PF maneuvers induced an early up-regulation of *ICAM1*, *CD44* and *ERK1* mRNAs after the first parabola (P1) and a delayed upregulation of *NFKB1, NFKBIA*, *NFKBIB,* and *FAK1* after the last parabola (P31). ICAM-1, VCAM-1 and CD44 protein levels were elevated, whereas the NF-κB subunit p-65 and annexin-A2 protein levels were reduced after the 31st parabola (P31). The *PRKCA*, *RAF1*, *BAX* mRNA were not changed and cleaved caspase-3 was not detectable in MDA-MB-231 cells exposed to PF maneuvers. Hyper-*g*-exposure of the cells elevated the expression of *CD44* and *NFKBIA* mRNAs, iRPM-exposure downregulated *ANXA2* and *BAX*, whereas VIB did not affect the TNBC cells. The early changes in ICAM-1 and VCAM-1 and the rapid decrease in the NF-κB subunit p-65 might be considered as fast-reacting, gravity-regulated and cell-protective mechanisms of TNBC cells exposed to altered gravity conditions. This data suggest a key role for the detected gravity-signaling elements in three-dimensional growth and metastasis.

## 1. Introduction

The GLOBOCAN statistics from 2018 showed that breast cancer is the most common cancer in women and the second leading cause of cancer deaths worldwide [1]. Breast cancer comprises multiple subtypes with characteristic histology, treatability and outcome [2]. Currently, seven molecular subtypes are known [3,4,5,6]: luminal A (estrogen receptor positive (ER+)/progesterone receptor positive (PR+)/androgen receptor positive (AR+)/human epidermal growth receptor 2 negative (HER2−)/antigen KI67 negative (KI67−)), luminal B (ER+/PR+/AR+/HER2−/KI67+), HER2 enriched (ER−/PR−/HER2+), molecular apocrine (ER−/PR−/AR+/HER2+/KI67+), basal-like/triple-negative (ER−/PR−/AR−/HER2−), normal breast-like (ER+/PR+/HER2−/KI67−) and claudin-low (ER−/PR−/HER2−). Survival of the patients is associated with prognostic factors like tumor size, hormone-receptor-profile and metastases at the time of diagnosis. An onset therapy is the surgical resection of the tumor tissue (lumpectomy or mastectomy). Tumor size and possible metastases to the sentinel lymph node are decisive for the choice of following therapy. Chemotherapy, radiation and anti-hormone therapy, targeted treatment against HER2 and anti-angiogenic therapy are often applied after surgery or in the case of advanced disease stages [7]. Despite the advanced therapy, it is estimated that about 626,679 deaths wordwide will occur from breast cancer per year [1]. Therefore, it is necessary to implement novel ideas to find drug targets and to test new treatment options. This study will focus on the well-described basal-like/triple-negative cancer (TNBC) and will study the TNBC cell line MDA-MB-231 exposed to short-term altered gravity conditions and vibration (VIB) [8]. Using altered gravity conditions in the field of cancer research is not typically the first method to search for target proteins [9], but more than 20 years ago it became clear that cells exposed to microgravity (µ*g*) opened an alternative view on cancer cells as they revealed numerous changes like an altered gene expression, protein synthesis and secretion. With the help of simulated microgravity (s-µ*g*) and mass spectrometry, we found novel protein targets involved in cancer [10,11,12,13]. A space journey and a longer stay on the International Space Station (ISS) provide altered gravity conditions, which are not found on Earth. On Earth, we can simulate µ*g* to some extent by applying rotating devices like a 2D or 3D clinostat, the National Aeronautics and Space Administration (NASA)-developed rotating wall vessel or a random positioning machine (RPM) [14]. Exposure of different cancer cells types to real (r-µ*g*) or simulated (s-)µ*g*-conditions influences various biological processes, which are of importance in cancer research. Other researchers reported about changes in the morphological phenotype, growth behavior, gene expression, protein content and release, cytoskeletal reorganization, extracellular matrix composition, cell adhesion and others [14,15]. One important finding was that in r- and s-µ*g* various cell types form three-dimensional (3D) aggregates also called multicellular spheroids (MCS), resembling the in vivo situation of tumors much closer than conventional cell cultures [14]. These MCS are of great interest in cancer research to test drugs and to find new treatment targets [16]. An activation of nuclear factor “kappa-light-chain-enhancer” of activated B-cells (NF-κB), a proinflammatory transcription factor, was often detected in breast cancer [17]. The “inhibitor of κB” (IκB) proteins include IκBα, IκBβ, IκBγ, IκBε, and others [18]. Among them, IκBα, IκBβ and IκBε are the most important regulators of NF-κB and are of high interest in cancer research and µ*g*-based research. Interestingly, the NF-κB-signaling pathway was significantly altered in FTC-133 cancer cells [19] as well as in MCF-7 breast cancer cells [20] exposed to s-µ*g* when MCS were formed. Grosse et al. described an increase in NF-κB p65 protein, when cells were exposed to s-µ*g* on an RPM [19]. This discovery was in concert with findings by Kopp et al., who described an activation and increase in NF-κB and associated molecules in MCF-7 cells exposed to the RPM [20]. Through drug-initiated NF-κB inhibition, they were able to reduce the formation of MCS. As it is not clear when NF-κB signaling is triggered during MCS formation, we exposed MDA-MB-231 breast cancer cells to r-µ*g* during a parabolic flight campaign (PFC).

The principal aim of this study was, first, to investigate the early phases of r-µ*g* achieved by PF maneuvers on TNBC cells and to test whether there is a link between factors of apoptosis, changes in NF-κB signaling and cell adhesion. The second aim was to study VIB and hyper-*g* (1.8 *g*) effects on the MDA-MB-231 cells. In a third approach, we exposed the MDA-MD-231 cells to an incubator RPM (iRPM) for 2 h to compare the effects of s-µ*g* with those from r-µ*g*. Afterwards, we performed quantitative polymerase chain reaction (qPCR) focusing on genes involved in NF-κB signaling, cell adhesion and MAPK signaling as well as apoptosis. Furthermore, the molecular biological results were finally evaluated by STRING (Search Tool for the Retrieval of Interacting Genes/Proteins) analyses to visualize the mutual regulation and interactions of genes and proteins examined in this study.

## 2. Results

### 2.1. Viability Staining

To examine whether the MDA-MB-231 cells were viable after the VIB-, hyper-*g*-, and iRPM-exposure, a terminal deoxynucleotidyl transferase dUTP nick end labeling (TUNEL) assay for the detection of DNA fragmentation was performed to measure the amount of apoptotic cells (Figure 1). Comparing ground controls at Earth’s normal gravity (1 *g*), VIB, 1.8 *g* hyper-*g* (comparable to the hyper-*g* exposure on the PFC), and iRPM cell samples, the cytoplasm was evenly stained green, while the nucleus showed no green staining. In contrast, the positive control, which was treated with DNase prior to the staining procedure, presents an intensive green staining of the nucleus. This finding shows, that altered gravity conditions or VIB did not induce apoptosis in MDA-MB-231 cells (Figure 1).

### 2.2. Nuclear Factor ”Kappa-Light-Chain-Enhancer” of Activated B-Cells (NF-κB) Signaling Factors in Triple-Negative Breast Cancer Cells during Altered Gravity Conditions

To verify the influence of PF maneuvers on factors involved in the NF-κB-signaling pathway, the gene expression pattern of NF-κB signaling factors was determined in MDA-MB-231 cells (Figure 2). As a PF comprises different conditions beside microgravity like hyper-*g* (1.8 *g*), and VIB, these factors had been additionally tested to isolate the effects derived from the exposure to µ*g* (Figure 3).

The major factors of the NF-κB-signaling pathway are NF-κB subunits p105/50, p100/52 and p65 (*NFKB3*, *RELA*) of which their corresponding genes are *NFKB1*, *NFKB2* and *NFKB3*.

While the gene expression of *NFKB1* (P31, up-regulation) (Figure 2A) and *NFKB3*, (*RELA;* P1, up-regulation, Figure 2C) are significantly changed after the PF conditions, VIB-, 1.8 *g*- and iRPM-exposure had no significant impact on these genes. The *NFKB2* mRNA was not altered in any experimental condition (Figure 2B, Figure 3C,D). In contrast, the Western blot analyses of NF-κB p65 protein presented a significant reduction after P1 and P31 (Figure 2D).

The NF-κB-signaling pathway is modulated by its inhibitors NF-κB-inhibitor-alpha, -beta and -epsilon (*NFKBIA, NFKBIB, NFKBIE*) and the NF-κB essential modulator (*IKBKG*, *NEMO*). In the case of *NFKBIA*, *NFKBIB* and *NFKBIE* (Figure 2E,G,H) gene expression, a significant upregulation was only found for *NFKBIA* and *NFKBIB* after P31 compared to their corresponding controls. The *NFKBIA* mRNA was differentially expressed by hyper-*g* (Figure 3G). Protein analyses revealed no significant change in IκBα and NEMO (Figure 2F,J). The *NFKBIE* and *IKBKG* gene expressions (Figure 2H,I) were not altered in any of the experimental conditions (Figure 3K–N).

### 2.3. Expression of Factors Belonging to the Biological Process of Apoptosis

Caspase 3 is a major factor in apoptosis [21]. Gene expression of *CASP3* was significantly upregulated after P1 and P31 (Figure 4A) while being not regulated after exposure to vibration and the RPM (Figure 5A,B). Measuring the cleaved caspase-3 protein by Western blot analysis and could not detect any active caspase-3, whereas the positive control colon cancer cells CX+ exerted a strong positivity [21] (Figure 4B).

The gene expression of *ANXA1* (Figure 4C, Figure 5C,D) was not significantly changed during any of the experimental procedures. *ANXA2* mRNA expression showed a similar behavior (Figure 4D, Figure 5E) except a downregulation after 2h of RPM exposure (Figure 5F). ANXA2 (annexin A2) protein (Figure 4E) was first significantly increased after P1 and then re-adapted after P31, which is in agreement with the data obtained after a two-hour iRPM-exposure of the MDA-MB-231 cells (Figure 5F). In addition, the *BAX* mRNA was not differentially regulated (Figure 4F, Figure 5I,J), but *BCL2* mRNA was reduced at both time points (Figure 4G). Finally, the *BAX* mRNA was not altered by VIB and hyper-*g*, but signicantly reduced in MDA-MB-231 cells exposed to the iRPM (Figure 5G,H).

### 2.4. Regulation of Cell Adhesion Molecules in Triple-Negative Breast Cancer (TNBC) Cells

The *ICAM1* gene expression was significantly upregulated after one parabola. Although our data seem to hint towards an additional upregulation after 31 parabolas, the high variability of the measurements interfered with the statistical analysis (Figure 6A). ICAM1 protein was significantly reduced after the first parabola and increased after 31 parabolas (Figure 6B). We did not observe changes in the *VCAM1* gene expression (Figure 6C), while VCAM-1 protein levels were significantly elevated after one and 31 parabolas compared to static 1 *g*-controls (Figure 6D).

*SPP1* gene expression was found to be significantly enhanced compared to the levels after one parabola, but not to 1 *g*-controls (Figure 6E). Osteopontin protein expression on the other hand was significantly downregulated compared to static controls after both one and 31 parabolas (Figure 6F).

Finally, the *CD44* gene expression showed a constant increase over the course of the experiment, with significant rises after one parabola vs. 1 *g*-controls and 31 parabolas vs. 1 *g*-controls and levels after one parabola (Figure 6G). CD44 protein levels followed the same tendency with significant increases compared to static controls for both one and 31 parabola samples (Figure 6H).

*ICAM1*, *VCAM1*, and *SPP1* were not significantly regulated after VIB, hyper-*g*- or iRPM-exposure (Figure 7A–F). The *CD44* gene expression, on the other hand, was significantly enhanced under conditions of 1.8 *g* hyper-*g,* but remained unchanged by VIB- and iRPM-exposure (Figure 7G,H).

### 2.5. Factors of the Mitogen-Activated Protein Kinase (MAPK) Signaling Pathway Known to be Involved in Cancer Progression and Metastasis

First, we determined the gene expression of *PRKCA* (protein kinase C alpha, PKC α). The *PRKCA* mRNA was not significantly upregulated after the 31st parabola compared with corresponding 1 *g*-samples (Figure 8A).

Then we focused on *RAF1* (RAF proto-oncogene serine/threonine-protein kinase). *RAF1* remained unchanged by PF maneuvers (Figure 8B). Afterwards, we measured the *ERK1* (extracellular signal-regulated kinase 1) (Figure 8C) and *ERK2* (extracellular signal-regulated kinase 2) (Figure 8D) gene expression. The *ERK1* gene expression was significantly up-regulated after the first parabola compared to 1 *g*-samples (Figure 8C). The *ERK2* mRNA was significantly elevated after the 31st parabola compared with P1 r-µ*g*-samples (Figure 8D). There is a direct interaction between *ERK1*,*2* and *FAK1* (focal adhesion kinase 1 also known as PTK2 (protein tyrosine kinase 2)). The *FAK1* mRNA was significantly up-regulated after P31 compared to 1 *g*-samples (Figure 8E).

In a next step we measured the *MAPK8* (mitogen-activated protein kinase 8) mRNA also called *JNK1* (c-Jun N-Terminal Protein Kinase 1) (Figure 8F). *JNK1* was down-regulated after P1 and was not significantly changed after P31, indicating a rapid short-term effect of microgravity.

### 2.6. Search Tool for the Retrieval of Interacting Genes/Proteins (STRING) Analysis

The various genes analyzed by qPCR were investigated in regard to their possible interaction and mutual expression dependence. A STRING/EMBL analysis of these items represented in molecule action mode is shown in Figure 9. It can be seen that the nuclear factors, whose expression was analyzed, are regulating each other very strongly, while the remaining items form a loose network. *RELA*, *NFKB1* and *NFKB2*, which represent three of the five members of the transcription factor NF-κB family are strictly controlled at various levels [22]. Their activity is regulated by interaction with each other, with inhibitors such as NFKBIA, NFKBIB, NFKBIE and with kinases such as IKBKB (NEMO). The regulation occurs at various levels including the gene expression level. For example, NFKB1 and RELA form protein heterodimers and are also associated in the regulation of their expression [23]. Moreover, *RELA* and *VCAM1* are upregulated after an injury of an artery, while *NFKBIA* is down-regulated [24] and if annexin A1 is upregulated, it forms a complex with IKFKG that activates NFKB [25]. On the other hand, the nuclear factors control the transcription of other genes. A target of RELA is *ICAM* expression, which is also regulated by osteopontin [26,27]. Osteopontin is co-expressed with *CD44* [28], which co-localizes with annexin A2 [29]. Moreover, there are links between NFKB1 and the expression of *CASP3* [30,31]. *CASP3* and *BAX* are upregulated during apoptosis, while *BCL2* is downregulated [32]. After the PF maneuvers, we measured increases in *CASP3* and a reduction of *BCL2* mRNAs, but no changes in their protein products and no signs of apoptosis. In addition, we detected an upregulation of *ERK1* after P1 compared to 1 *g*, *ERK2* after P31 compared to P1 samples, and *FAK1* after P31 compared to 1 *g*, whereas PRKCA and RAF1 remained unchanged in short-term µ*g*. Moreover, the *JNK1* gene expression was down-regulated after the first parabola and unaltered after P31 compared to 1 *g*.

## 3. Discussion

The basal-like or triple-negative form of breast cancer refers to tumors not expressing the *ER*, *PR*, *AR* and *HER2* genes [1]. This breast cancer type is very heterogeneous, which complicates treatment. The MDA-MB-231 cell line was isolated from a pleural effusion of a patient with invasive ductal carcinoma and is ER-, PR-, and E-cadherin negative and expresses mutated p53. Microarray profiling revealed that the MDA-MB-231 cell genome clusters with the basal subtype of breast cancer. In addition, MDA-MB-231 cell lack HER2 and thus this cell line is a good model of TNBC [8,33,34].

Gravity is permanently influencing the human body and all life on Earth. Gravitational unloading results in enormous changes in the organism, organs, tissues and cells [35,36].

Microgravity induces a large number of changes in thyroid and breast cancer cells, such as alterations of the cytoskeleton, extracellular matrix, focal adhesion, growth behavior, differentiation, proliferation, cell adhesion or an increased apoptosis [15,20,37,38,39,40].

We recently exposed MCF-7 breast cancer cells to the RPM and detected 3D MCS formation within 24 h [20]. The pathway analysis of 47 examined genes proposed that NF-κB variants are involved in the formation of MCS [20]. This finding was in agreement with data obtained earlier from thyroid cancer studies [19]. Poorly differentiated FTC-133 follicular thyroid cancer cells cultured on an RPM for 24 h showed higher levels of NF-κB-p65 protein and apoptosis than 1 *g*-controls [19]. Both studies indicated an involvement of NF-κB in 3D growth. No information exists about the early phases when cells are exposed to µ*g*.

We had the opportunity to attend the 29th Deutsches Zentrum für Luft- und Raumfahrt (DLR) PF mission in Bordeaux Merignac, France, and to study cells in r-µ*g* (Available online: https://www.dlr.de/rd/desktopdefault.aspx/tabid-2285/3423_read-47055/; https://www.dlr.de/rd/desktopdefault.aspx/tabid-2285/3423_read-50372/). We investigated MDA-MB-231 cells in r-µ*g* during PF maneuvers. The cells were fixed with RNA*later* after the first and the 31st parabola, keeping in mind that this last time point represents a mixture of alternated acceleration phases. Postflight we focused on NF-κB signaling, cell adhesion, and apoptosis. Furthermore, cells exposed to an iRPM, hyper-*g* cultures (1.8 *g*) and VIB samples were examined to evaluate the impact of these factors on the cells.

### 3.1. Influence of Altered Gravity Conditions and Vibration on Cell Survival

Apoptosis is involved in the pathogenesis of many diseases including among others ischemia, autoimmune and neurodegenerative diseases, as well as in tumor response to chemotherapy and/or radiotherapy [41].

It is known from earlier studies that cancer cells will become apoptotic when exposed to r-µ*g* and s-µ*g* and that apoptosis is involved in cell detachment and formation of spheroids [19]. Therefore, we investigated *CASP3*, *ANXA1*, *ANXA2*, *BAX* and *BCL2* mRNA expression patterns.

In our short-term experiments, after a 2-h exposure to s-µ*g*, hyper-*g* or vibration, TUNEL staining of TNBC cells revealed no signs of apoptosis and 100% viable cells. The gene expression of *CASP3* was elevated after the first and 31st parabola, but no uncleaved caspase-3 was detectable and no detectable cleavage of caspase-3 occurred. The *CASP3* gene expression was not changed in VIB-, hyper-*g*- and s-µ*g*-samples. This is in agreement with earlier findings obtained with endothelial cells exposed to PF maneuvers [42]. Furthermore, the gene expression of *CASP3* and *ANXA2* was reduced in endothelial cells exposed to a two-hour period of VIB or hyper-*g* conditions [43]. In MDA-MB-231 cells *ANXA1* mRNAs were not changed during the parabolic flight, hyper-*g*-, s-µ*g*- and VIB-exposure. However, ANXA2 protein after PFC as well as *ANXA2* mRNA expression after s-µ*g* exposure was significantly altered. Thus, µ*g* has an impact on cell survival that is highly dependent on the cell type as well as on the duration of exposure.

### 3.2. Impact of Real Microgravity on NF-κB Signalling in TNBC

NF-κB is an interesting factor because it is associated with spaceflight-related health problems. Activation of NF-κB is frequently observed in breast cancer as well. An *NFKB3* overexpression indicates increased aggressiveness of breast cancer and a poor prognosis [44]. In addition, NF-κB is involved in endocrine therapy resistance [45]. Activation of NF-κB promotes the survival of tumor cells. Several gene products that negatively regulate apoptosis in tumor cells are controlled by NF-κB activation.

The NF-κB proteins include the different variants NF-κB-p50, -p52 and -p65, which are encoded by the gene loci *NFKB1*, *2* and *3* [45]. These proteins are bound and inhibited by IκB proteins. The effectors and inhibitors are activated by external triggers and, thus, interesting in µ*g-*based research.

Human adherent cells exposed to s-µ*g* on a RPM showed elevated levels of NF-κB p65 protein compared with 1 *g*-controls, a result found earlier in endothelial cells and in FTC-133 follicular thyroid cancer cells [19,46].

NF-κB exerts several transcriptional regulatory functions important for programmed cell death [47] and is inactivated by binding to IκB (inhibitor of NFκB).

The *NFKB1* (Nuclear Factor Kappa B Subunit 1) mRNA was significantly elevated after the 31st parabola compared to 1 *g*, no changes were measured for the *NFKB2* (Nuclear Factor Kappa B Subunit 2) mRNA at each time point and an upregulated gene was found for *NFKB3* (*RELA* Proto-Oncogene (V-Rel Avian Reticuloendotheliosis Viral Oncogene Homolog A), Nuclear Factor NF-Kappa-B P65 Subunit) after the first parabola. *NFKB3* mRNA was unaltered after the last parabola (P31). Despite this result, the corresponding protein NF-κB p65 was down-regulated after the first and 31st parabola. This counterregulatory effect was not due to VIB or hyper-*g* occurring during the PF (Figure 3E,F), as the *NFKB3* gene expression was unchanged in VIB and 1.8 *g* samples. The normalization of *NFKB3* after 31 parabolas and the down-regulation of NF-κB p65 protein in short-term r-µ*g* (after one parabola, that means 22 s of r-µ*g*) is in agreement with earlier microarray data obtained from activated T cells cultured in space, which showed a suppressed expression of c-REL and NF-κB gene targets after 1.5 h [48]. In addition, a four-hour exposure on the RPM of activated T cells revealed a suppressed expression of NF-κB gene targets [49].

In many cell types, NF-κB dimers are located in the cytoplasm in an inactive form through association with any of several IκB inhibitor proteins (IκBα, -β, -ε, -γ, p105 and p100) [50].

The NF-κB signaling pathway is mainly regulated by inhibitor κB (IκB) proteins and the IκB kinase complex through two major pathways: the canonical and non-canonical NF-κB pathways [50]. We measured significantly elevated *NFKBIA* and *NFKBIB* mRNAs after the 31st parabola, whereas the *NFKBIE* and *IKBKG* gene expression were not altered by r-µ*g* (Figure 2). VIB did not affect the family of cellular IκB proteins inhibiting the NF-κB transcription factors, but hyper-*g* elevated the *NFKBIA* mRNA expression indicating that the measured elevation may be due to the hyper-*g* phase of the parabola, because an iRPM-exposure of the MDA-MB-231 cells demonstrated no changes in NF-κB signaling. On the iRPM all investigated genes remained stably expressed. The RPM aims to simulate near weightlessness, but one should take note of increased fluid movements—and thus shear forces—occurring in the flasks,when the device is used for cell cultures experiments [51]. First, it is important to place the sample as close as possible to the center of rotation in order to minimize residual g-artifacts and second, the cell culture flasks have to be completely filled with medium without air bubbles to minimize shear stress. The RPM is used to prepare spaceflight missions and for tissue engineering purposes. It has proven to be a useful device for long-term cultures of cancer cells and benign cells [14]. Postflight data revealed that genes and proteins involved in the regulation of thyroid cancer cell proliferation and metastasis were similarly regulated under RPM and spaceflight conditions [52].

### 3.3. Parabolic Flight Maneuvers Changed Cell Adhesion Factors 

It is known that s-µ*g* conditions induced changes in the cytoskeleton, ECM, and focal adhesion factors in various cell types [53,54]. Long-term RPM-exposure of human endothelial cells induced the formation of 3D tubular structures and spheroids [55]. In parallel, secretion of the factors ICAM-1, and VCAM-1 were both increased, when the influence of gravity is minimized for 35 days [55]. In MDA-MB-231 cells exposed to short-term µ*g* on a parabolic flight, an increased synthesis of ICAM-1 protein was found after the P31. A similar result was measured for VCAM-1 protein as given in Figure 6.

The adhesion molecules ICAM-1 and VCAM-1 are mediating the cell adhesion of cancer cells, lymphocytes and other cells to the vascular endothelium [56]. Moreover, ICAM-1 is involved in angiogenesis and is able to increase the survival of microvessels [57].

Primary human macrophages differentiated from monocytes exposed to 11 days r-µ*g* in space on the ISS revealed a reduced surface expression of ICAM-1, defucosylation of surface proteins and an altered metabolite spectrum [58]. Another study examined the ICAM-1 protein synthesis and *ICAM1* gene expression in cells of the monocyte/macrophage system exposed to r- and s-µ*g* obtained during 2D clinostat, parabolic flight, sounding rocket, and orbital experiments [59]. Murine BV-2 microglial cells showed a downregulated *ICAM1* expression, when exposed to a 2D clinostat and a rapid and reversible downregulation in the µ*g*-phase of PF maneuvers [59].

Interestingly, an elevated *ICAM1* mRNA was measured in macrophage-like differentiated human U937 cells during the µ*g*-phase of PFs. In non-differentiated U937 cells, no effect of µ*g* was observed [59].

In addition, we studied the early effects of hyper-*g*, and VIB comparable to the conditions that occur during PFs on the gene expression of *ICAM1* and *VCAM1*. Both vibration and hypergravity had no effect on *VCAM1* expression. Hyper-*g* did not change the *ICAM1* mRNA expression, while VIB-exposure of the MDA-MB-231 cells revealed a non-significant result due to the high variation. Furthermore, the *SPP1* gene expression was not altered by VIB and hyper-*g*, whereas *CD44* was unchanged by VIB, but significantly elevated by hyper-*g*. Therefore, it can be assumed that the elevated *CD44* gene and protein expression after P31 (Figure 6G,H) is mainly due to hyper-*g* (Figure 7G). Comparable results were obtained when poorly differentiated ML-1 follicular thyroid cancer cells were investigated during a PFC. The *SPP1* mRNA was significantly elevated after P31 [60]. The *SPP1* gene expression was not changed by VIB and hyper-*g* in ML1 follicular thyroid cancer cells [60]. Corresponding data were obtained when human chondrocytes were studied during PF maneuvers [61]. No significant changes in the gene expression levels were observed during VIB and hyper-*g* experiments [61].

In a next step, we focused on the cell-surface glycoprotein CD44 antigen, which is involved in cell adhesion, migration and cell-cell interactions. The CD44 is a receptor for hyaluronic acid and interacts with osteopontin. The mRNA expression of *CD44* and the corresponding protein content were both significantly elevated after P1 and P31 compared to 1 *g*. Similar results were obtained earlier, when cells were exposed to a NASA rotary cell culture system (RCCS) grown as 3D spheroids and were CD44-positive [62]. The CD44-positivity of the cells grown in 3D MCS was determined by immunocytochemistry and was elevated in bladder cancer, prostate cancer, and glioma cell lines compared with 1 *g*-cultures [62].

Rat osteoblasts cultured for 4 or 5 days aboard the Space Shuttle and solubilized during spaceflight revealed strongly elevated *CD44* mRNA levels in the flight cultures [63]. In addition, FTC-133 follicular thyroid cancer cells exposed to an RPM for 24 h expressed a *CD44* mRNA which was significantly up-regulated in adherent cells but not significantly altered in MCS [19]. In addition, an increase in *SPP1* mRNA was measured in adherent FTC-133 cells cultured on the RPM. *CD44* can also interact with *SPP1*. The *SPP1* mRNA was elevated in MDA-MB-231 cells after P31 compared with P1 and 1 *g*-samples. The synthesis of the protein was reduced during the PF maneuvers (Figure 6F), whereas both hyper*-g* and VIB stress had no effect on *SPP1*. These results are in concert with data obtained from thyroid cancer cells exposed to the RPM. Simulated µ*g* reduced the amount of osteopontin in adherent cells and MCS. Based on the role of osteopontin as a mediator of cell-matrix adhesion and communication, it is influencing tumorigenesis and invasion [64]. Osteopontin seems to be involved in motility regulation by interaction with CD44 in colon cancer cells, which suggests a role for osteopontin in cancer progression [65]. Osteopontin is a potential cancer biomarker [66] and is involved in biological processes such as cell proliferation, survival, angiogenesis, progression and metastasis [66].

Another study demonstrated that in MDA-MB-231 cells, the inhibition of NF-kB via the chemical compound Bay-11-7082 results in a CD44 suppression [67]. The NF-kB inhibition and subsequent CD44 suppression reduced the cell proliferation and invasiveness of breast cancer cells. In contrast in µ*g*, the cells reacted with a down-regulation of NF-kB p65 protein and an increase in CD44, a finding which has to be investigated in long-term µ*g*-studies in the future.

### 3.4. Interaction Network of Selected Genes Evaluated by STRING Analysis

The interaction between CD44 and osteopontin as a potential basis for cancer progression and metastasis formation is known for a long time [68]. In this study, the up-regulation of the *CD44* mRNA seems to be induced by the hyper-*g* phases of the PF maneuvers, whereas *SPP1* is mainly elevated by r-µ*g.* Interestingly, the synthesis of osteopontin is reduced, a finding only observed in r-µ*g*.

Furthermore, osteopontin regulates the *ICAM-1* and *VEGFA* expression mainly in triple-negative/basal-like breast cancer, supporting its role in tumor progression in TNBC [27]. Osteopontin protein was reduced in P31 samples, whereas the ICAM-1 protein synthesis was elevated after 31 parabolas which may allow for both proteins to conclude a counterregulatory interaction mechanism in short-term µ*g.*

Another group demonstrated that purified native OPN induces NF-κB activation and NF-κB-dependent ICAM-1 expression in breast cancer cells [69]. We measured a reduced amount of NF-κB after P1 and P31 which may be due to the reduced osteopontin content (Figure 2D, Figure 6F).

Both cell adhesion molecules ICAM-1 and VCAM-1 are increased in patients with advanced breast cancer and the increase in VCAM-1 is of prognostic significance [70]. In our study the VCAM-1 protein synthesis was elevated after P1 and P31. *VCAM1* is one of five genes (*CXCL12*, *MMP2*, *MMP11*, *VCAM1*, and *MME*), which were associated with tumor progression, angiogenesis, and metastasis [71].

There is also an interaction between *CD44* and *ANXA2*. *ANXA1* together with *ANXA2* are both associated with the aggressive behavior of TNBC [72]. The prognostic impact of *ANXA1* relies on a high *ANXA2* expression and both are preferential for TNBC [72]. The MDA-MB-231 cells exhibited a high expression of *ANXA1* and *ANXA2. ANXA1* remained stable under all gravity conditions and VIB. Both, r- and s-µ*g* induced a reduction of ANXA2 protein after 31 parabolas and *ANXA2* expression after 2h RPM exposure. This response seems to be due to altered gravity.

ANXA1 is known to constitutively activate NF-κB in breast cancer cells by interacting with the IKK complex [25], an interaction that might not to be relevant in µ*g*.

*ANXA2* is upregulated in many cancer types and is involved in cancer cell migration, adhesion, invasion, and metastasis [73]. Intracellular annexin A2 regulates NF-κB signaling by binding to the p50 subunit in a calcium-independent manner [73]. The ANXA2-p50 complex translocated into the nucleus [73]. After the first parabola annexin A2 protein synthesis was elevated and then after the 31st parabola reduced. The *NFKB1* mRNA was upregulated after P31, but was not altered under all other experimental conditions (Figure 2A, Figure 3A,B).

*MAPK3* (*ERK1*) was upregulated after the first parabola and is interacting with *RELA*, *CASP3* and *BCL2*. It is known that NF-κB inhibits ERK activation to enhance cell survival during the development of tumor adaptive radioresistance in breast cancer cells [74]. In addition, ERK1/2 play a key role in controlling the BCL2-regulated, cell-intrinsic apoptotic pathway [75]. Other interactions are found for OPN, FAK and ERK1/2 as well as RELA. Integrin αvβ3 binding with OPN mediates the signaling pathways of FAK, ERK1/2, and NF-κB to activate cellular migration [66], which is important for the dissemination of cancer cells to distant tissues.

### 3.5. MAPK Signaling Factors Involved in Cancer Progression and Metastasis

We investigated further factors known to enhance cancer growth and spreading. The protein kinase C alpha (PKCα) is implicated in cancer progression and associated with a poor prognosis in breast cancer patients [76]. There is evidence that PKCα is a key regulator of migration and invasion in endocrine resistant ER+ breast cancer and basal A TNBC, but not in other subtypes such as endocrine sensitive ER+ [76]. In this study, we found a slight increase in *PRKCA* in TNBP exposed to short-term µ*g* after 31 parabolas. This result was not significant, and thus it may be speculated that *PRKCA* is upregulated later, when the cell detachment starts as well as the 3D aggregation of breast cancer cells [40]. In addition, *PRKCA* is also a key candidate gene in melanoma metastasis [77]. An important step for metastasis is that protein kinase C alpha activates RAF-1 by direct phosphorylation [78].

The biological process of the epithelial-mesenchymal transition (EMT) is known to increase migration and spreading of cancer cells, progression of the cell cycle, and resistance to apoptosis and chemotherapy. It supports tumor progression. One important signaling pathway involved in progression and metastasis is the MAPK pathway [79]. Therefore, we determined the *RAF1* mRNA expression. RAF1 works as a regulatory link between the membrane-associated Ras GTPases and the MAPK/ERK cascade and functions as a switch determining proliferation, differentiation, apoptosis, survival and oncogenic transformation of human cells [80]. RAF1 activation initiates a mitogen-activated protein kinase (MAPK) cascade [80]. The fact that it remained unaltered might explain that we did not find any apoptosis in TNBC exposed to short-term microgravity (Figure 8B). 

Furthermore, we measured the expression of *ERK1* and *ERK2.* The ERK subfamily consists of typical (ERK 1/2/5), and atypical (ERK 3/4/7/8) members. ERKs are involved in the regulation of EMT and are thus promoting tumor progression. The *ERK1* gene expression was significantly up-regulated after the first parabola compared to 1 *g*-samples, and *ERK 2* after the 31st parabola compared with P1 r-µ*g*-samples (Figure 8D). ERK1 seems to give an initial signal to start 3D aggregation in µ*g*.

Moreover, we focused on the focal adhesion kinase 1 (FAK1), which is also called PTK2 protein tyrosine kinase 2 (PTK2) and known to increase cancer cell migration and promote metastatic dissemination to distant sites [81]. Blocking of FAK revealed that breast cancer cells became less metastatic due to decreased mobility [81]. The *FAK1* mRNA was significantly up-regulated after P31 compared to 1 *g*-samples. This is an interesting result indicating its importance for 3D aggregation, breast cancer progression and invasion. Chan et al. [82] had demonstrated that the depletion of FAK induced the formation of active invadopodia and impaired invasive cell migration.

Finally, we measured the *MAPK8*/*JNK1* mRNA in our experimental µ*g*-dependent approach. The *JNK1* gene was down-regulated after the first parabola in the µ*g*-exposed TNBP cells.

It has been shown that *JNK1* promotes cell survival in Her2/neu-positive breast cancer [83]. Human studies have shown the relevance of JNK activation to various human cancers [84]. These kinases are involved in the prevention of malignant transformation via the induction of apoptosis and in promoting cell survival in established tumors [85]. In addition, there is a potential to monitor JNK activity as an early biomarker of response to chemotherapy [85]. Therefore, the decrease in *JNK1* mRNA expression seems to promote a following 3D formation of the TNBC cells in real µ*g-*conditions.

Taken together, TNBC cells exposed to short-term µ*g* obtained by PF maneuvers kept all signs of a more aggressive phenotype as elevations of ICAM1 and VCAM1 proteins occurred soon.

MDA-MB-231 cells exposed to short-term r-µ*g* were relatively stable to this external stressor, vibration and hyper-*g*.

In summary, an early up-regulation of *NFKB1* (P1), *NFKB3* (*RELA*) (P1), *ERK1* (P1), *ICAM1* (P1), *NFKBIA* (P31), *NFKBIB* (P31), *FAK1* (P31), *SPP1* (P31) and *CD44* (P1, P31) gene expression as well as a reduced protein content of NF-κB p65 and osteopontin were found after the PF maneuvers. *CD44* and *NFKBIA* were upregulated in hyper-*g,* showing that the hyper-*g*-phase seems to influence both factors. For all other genes, our data indicate that the microgravity phase is the driving factor of most of these changes in gene expression. These results are very important because apoptosis is needed for cell detachment together with an activation of NF-κB-p65 to form 3D growth (spheroids) of cancer cells when they were exposed to µ*g*. Changes in osteopontin protein suggest a role in survival, angiogenesis, invasion, and metastasis of TNBC cells.

## 4. Materials and Methods

### 4.1. Cell Culture

MDA-MB-231 cells were purchased from the American Type Culture Collection (ATCC^®^ HTB26™, Manassas, Virginia, USA)). They were cultured in RPMI 1640 (Life Technologies, Paisley, UK), 10 % FCS (Sigma Aldrich, Steinheim, Germany) and 1% penicillin/streptomycin (Life Technologies, New York, USA). The cells were cultured in vented T75 cm^2^ flasks (Sarstedt, Nümbrecht, Germany) and were split every 4–5 days to prevent confluence.

### 4.2. 29th. Parabolic Flight Campaign

The PFs were organized by the DLR in cooperation with Novespace, Bordeaux-Mérignac Airport, France. The cells were transferred on board of the Airbus A310 (Figure 10A) into a pre-warmed 37 °C incubator shortly before take-off and they were incubated at 37 °C for the whole time during the flight. The flight rack is shown in Figure 10B. Each parabola had an initial phase of hyper-*g* (1.8 *g*) for 22 s during pull-up and a final phase of hypergravity for 22 s during pull-out (Figure 10C). The two hyper-*g*-phases framed a 22-s long µ*g*-phase. The flight maneuver was repeated 31 times per flight day. During the PFs, the MDA-MB-231 cells were fixed with RNA*later* (Invitrogen by Thermo Fischer Scientific, Vilnius, Lithuania) at the end of the first parabola (P1) and the end of the last parabola (P31). The cells were cultured in T75cm^2^ cell culture flasks (Sarstedt, Nümbrecht, Germany ) with 10 mL medium in each flask. Each flask had a fixed three-way connector on the lid which was connected with 140 cm tubing to a 50 mL syringe filled with RNA*later*. The RNA*later* was injected manually into the flasks at the designated times. Additional MDA-MB-231 cells were incubated on ground to serve as static 1 *g*-controls. The ground control cells were fixed with RNA*later* in parallel to the samples on board of the flight. After landing, the medium and RNA*later* mixture was removed and replaced with 5 mL of fresh RNA*later*. The cells were harvested with a 25 cm scraper (Sarstedt, Nümbrecht, Germany) and stored suspended in RNA*later* at 4 °C in 15 mL tubes until RNA isolation [54].

### 4.3. Vibration Experiments

As a PF implements different stressors, vibration was tested on the cells using the Vibraplex device (DLR, Cologne, Germany) [61,86]. MDA-MB-231 cells were seeded on µ-Slide VI ibiTreat channel slides (Ibidi, Gräfelfing, Germany) for immunostaining and in T25 cm^2^ cell culture flasks for RNA isolation. The flasks and the slides were attached to the Vibraplex platform. The Vibraplex was transferred into a 37 °C incubator. Frequencies between 0.2 to 14 Hz were produced via the Vibraplex for a two-hour duration. These frequencies resemble the VIB produced during the PF. After the vibration procedure, the cells were fixed with either 4% PFA for immunostaining or RNA*later* for RNA isolation. Ground control slides and flasks were cultured in parallel for comparison.

### 4.4. Hyper-g Experiments

MDA-MB-231 cells were counted and seeded into µ-Slide VI ibiTreat channel slides (Ibidi, Gräfelfing, Germany) comparable to [87] for immunostaining. For RNA isolation T 75 cm^2^ flasks were used with a confluence of 90%. The flasks and the slides were fixed on swing-out gondolas inside the DLR multi-sample incubator centrifuge (MuSIC) located inside a 37 °C incubator. The centrifuge was rotating at a constant speed to produce 1.8 *g* for 2 h which is comparable to the hypergravity phases produced during the parabolic flight. The device designed by the DLR, Department of Gravitational Biology, is shown in Figure 10D. At the end of the run, the cells were fixed with either 4% PFA for immunostaining or RNA*later* for RNA isolation. All slides and cell culture flasks were randomly assigned to centrifugation or 1 *g*-ground controls.

### 4.5. Incubator Random Positioning Machine (iRPM)

The iRPM (Figure 10E) was designed and constructed by Prof. Jörg Sekler-Fachhochschule Nordwestschweiz. Details of the device were published in [88].

MDA-MB-231 cells were seeded in µ-Slide VI ibiTreat channel slides (Ibidi, Gräfelfing, Germany) for immunostaining and T25cm^2^ flasks for RNA isolation. The slides and the flasks were placed inside a 37 °C pre-warmed iRPM. At the end of the 2 h, the cells were fixed with either 4% PFA for immunostaining or RNA*later* for RNA isolation. The RNA was isolated as mentioned earlier. All the slides and flasks were put in comparison to 1 *g*-ground control.

### 4.6. RNA Isolation and Quantitative Polymerase Chain Reaction (qPCR)

All falcon tubes were centrifuged (2500× *g* for 10 min at 4 °C), followed by discarding the supernatant. The RNA was isolated afterwards using the RNeasy Mini Kit (Qiagen, Hilden, Germany) according to the manufacturer’s protocol. The quality of the RNA was evaluated with a spectrophotometer. Afterwards, RNA was converted to cDNA with the High Capacity cDNA reverse Transcription Kit according to the manufacturer’s protocol (Applied Biosystems, Darmstadt, Germany). 1 µg total RNA in 20 µL reaction mix was prepared as stock for qPCR. The primers were designed using Primer Blast (primer designing tool from NCBI).

A total volume of 13 µL SYBR green reaction mix (Applied Biosystems, Darmstadt, Germany) was pipetted in each well in a 96 well plate. 1 µL of cDNA was added to each reaction mix with a concentration of 100 µM forward and reverse primers. The 7500 Fast Real-Time PCR System (Applied Biosystems, Darmstadt, Germany) was used to determine the transcription level of targeted genes (see Table 1). The program consisted of an initial 20 s long holding stage of 95 °C followed by the cycling stage. The cycling stage consisted of 40 cycles of 3 s at 95 °C and 30 s at 60 °C. A melting curve was implemented at the end of each run to verify the primer specificity. The data was collected and analyzed by the ΔΔ*C*_T_ method. 18S rRNA and TBP were used as reference genes.

### 4.7. Western Blotting

Western blot analysis, gel electrophoresis, trans-blotting, and densitometry were carried out following routine protocols as described previously [21,54,61]. Following lysis and centrifugation, aliquots of 30 µ*g* were subjected to sodium dodecyl sulphate-polyacrylamide gel electrophoresis (SDS-PAGE) and Western blotting. The samples were collected at the end of the P1 and P31 and were compared to 1 *g* control samples. Each condition is represented with 5 samples with a total number of 15 samples for all the conditions per cell line. The samples were loaded onto Criterion XT 4–12% precast gels (Bio-Rad, Hercules, CA, USA) and run for 1 h at 150 V. Proteins were then transferred with a TurboBlot (Bio-Rad) (100 V, 30 min) to a PVDF membrane. Cofilin-1 was used as a loading control. Membranes were then blocked for 2 h in TBS-T containing 0.3% I Block (Applied Biosystems, Foster City, CA, USA). For the detection of the selected antigens (see Table 2), the membranes were incubated overnight at room temperature in TBS-T and 0.3% I Block solutions of the antibodies. Following three washing steps of 5 min, the membranes were incubated for a further 2 h at room temperature with the secondary antibody Horseradish peroxidase (HRP)-linked antibody (Cell Signaling Technology Inc., Danvers, MA, USA) diluted 1:4000 in TBS-T and 0.3% I-Block. The respective protein bands were visualized using Bio-Rad Clarity Western ECL (Bio-Rad) and images were captured with Image Quant LAS 4000 mini (GE Healthcare Life Science, Freiburg, Germany). Images of stained membranes were captured on Syngene PXi 4EZ image analysis system (Synoptics, Cambridge, UK) and analyzed using the ImageJ software for densitometric quantification of the respective bands and total protein load.

### 4.8. Terminal Deoxynucleotidyl Transferase dUTP Nick End Labeling (TUNEL) Staining

The MDA-MB-231 cells were cultured in µ-Slide VI 0.4 ibiTreat Ibidi slides (IBIDI GmbH, Martinsried, Planegg, Germany), exposed to VIB, hyper-*g* and iRPM and subsequently collected for the detection of apoptosis. The method was published earlier in Lützenberg et al. [89].

TUNEL staining was performed according to the manufacturer’s recommendation (Thermo Fisher Scientific, Waltham, Massachusetts, USA; Click-iT TUNEL Alexa Fluor 488 (cat# C10245)).

The stained cell samples (VIB, hyper-*g*, iRPM and corresponding static 1 *g*-controls) were examined utilizing a Leica DM 2000 microscope equipped with an objective with a calibrated magnification of 400× and connected to an external light source, Leica EL 6000 (Leica Microsystems GmbH, Wetzlar, Germany). To obtain positive controls the cells were treated with DNAase before the TUNEL staining.

### 4.9. STRING Analysis

Interactions between proteins were determined using the STRING 10 platform [90]. For each protein, the UniProtKB entry number was inserted in the input form “multiple proteins” and “Homo sapiens” was selected as organism. The resulting network view was downloaded in the molecular action view showing lines between interacting proteins and genes [91].

### 4.10. Statistical Analyses

GraphPad prism 7.01 (GraphPad Software, Inc., California, USA) was used to analyze the data. The nonparametric Mann-Whitney U test was used as a statistical test of significance. The difference between groups was considered significant when the *p*-value was less than 0.05 (* *p* < 0.05).

## 5. Conclusions

Short-term r-µ*g* produced by PF maneuvers induced the gene expression of cell adhesion molecules in triple-negative breast cancer cells. This finding is in agreement with long-term µ*g* (s- and r-µ*g*) data with other cell types grown on the NASA-developed high-aspect ratio vessel (HARV) or on the Space Shuttle in space for 4–5 days [62,63,92]. The CD44 upregulation in the r-µ*g*- and hyper-*g-*cultures may be involved in the compensative regulation to counteract cellular apoptosis occurring in µ*g* [93,94]. Cell adhesion molecules and factors of the MAPK pathway are involved in the adaptive response to perturbation of mechanical stress under short-term real microgravity. Overall, our study suggests that a fine balance between NF-κB-p65 and osteopontin gene dosage is required to regulate metastasis, survival and angiogenesis of TNBC cells.

## Figures and Tables

**Figure 1 ijms-20-05730-f001:**
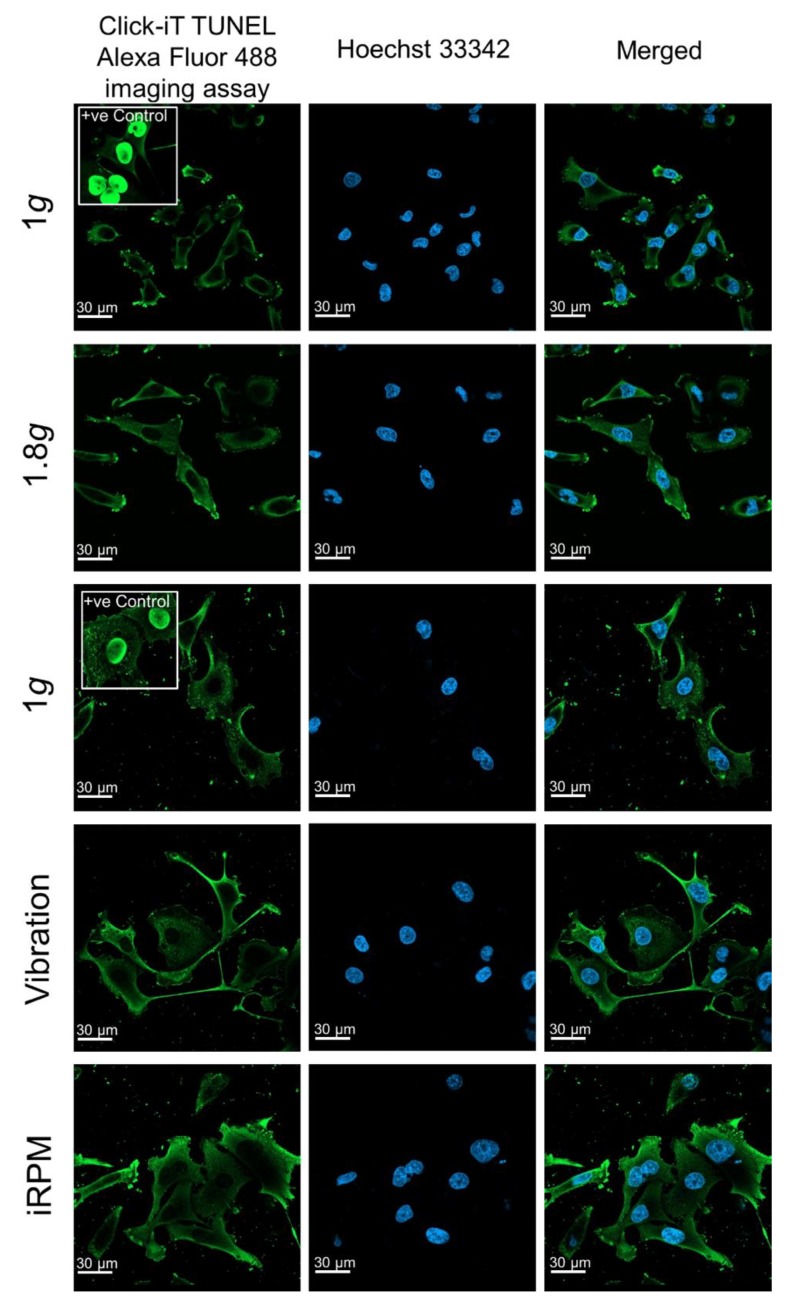
Click-IT terminal deoxynucleotidyl transferase dUTP nick end labeling (TUNEL) assay performed on MDA-MB-231 cells exposed to 1 *g*, 1.8 *g* hyper-*g*, vibration (VIB) and the incubator random positioning machine (iRPM). Green staining indicates free fluorophores in the cytoplasm in all images with the exception of the positive control. In the positive control, samples have been pretreated with DNase to induce DNA fragmentation, which is visualized by an enrichment of the fluorophores in the nucleus. Blue staining highlights the cells’ nuclei. Green stained nuclei present apoptotic cells as shown in the inserted positive controls. None of the applied experimental approaches had induced apoptosis in the cells.

**Figure 2 ijms-20-05730-f002:**
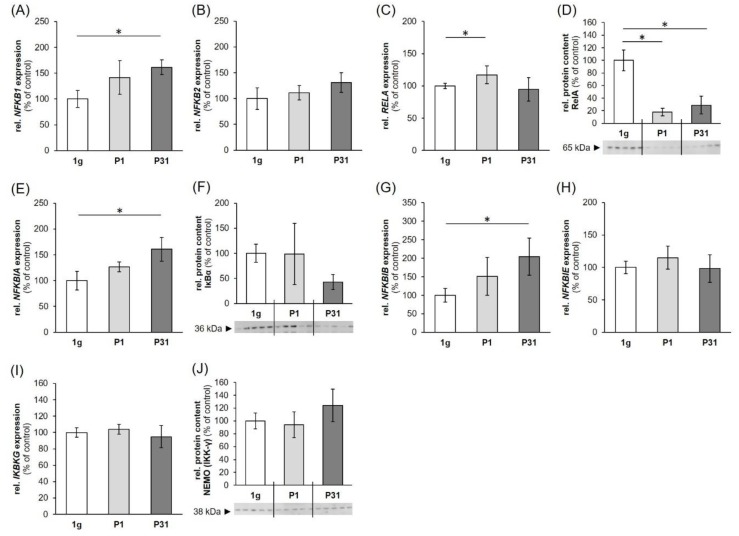
Influence of short-term microgravity on the gene expression: (**A**) *NFKB1*, (**B**) *NFKB2*, (**C**) *RELA*, (**E**) *NFKBIA*, (**G**) *NFKBIB*, (**H**) *NFKBIE*, (**I**) *IKBKG* and protein content: (**D**) RelA, (**F**) IκBα (**J**) NEMO; of NF-κB signaling factors. *n* = 5; The data are given as mean ± standard deviation. * *p* < 0.05 vs. 1 *g-control*.

**Figure 3 ijms-20-05730-f003:**
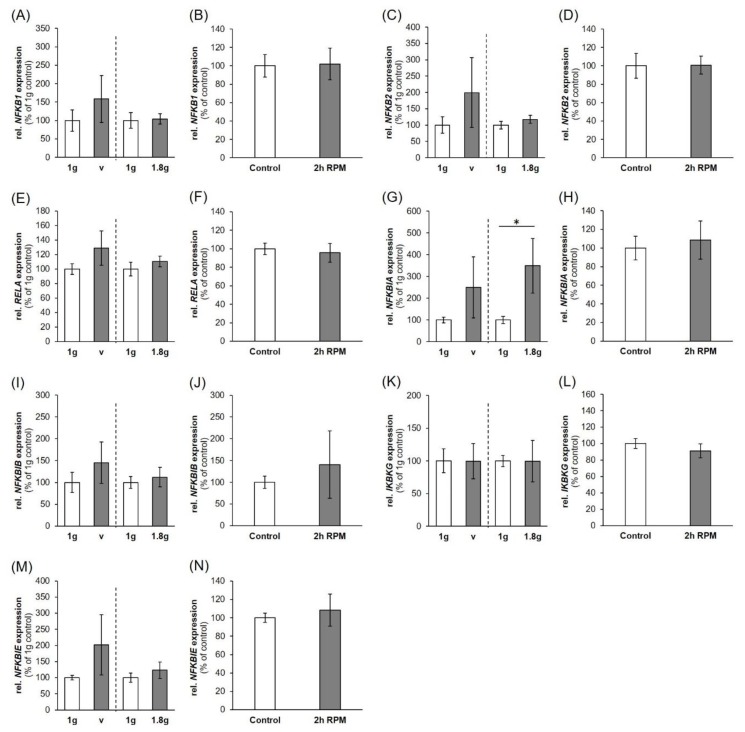
Influence of VIB (V), hyper-*g* and iRPM-exposure on the gene expression of NF-κB signaling factors: (**A**,**B**) *NFKB1*, (**C**,**D**) *NFKB2*, (**E**,**F**) *RELA*, (**G**,**H**) *NFKBIA*, (**I**,**J**) *NFKBIB*, (**K**,**L**) *IKBKG*, (**M**,**N**) *NFKBIE*. *n* = 5. The data are given as mean ± standard deviation. * *p* < 0.05 vs. corresponding 1 *g*-controls. The dashed vertical line seperates two independent experiments.

**Figure 4 ijms-20-05730-f004:**
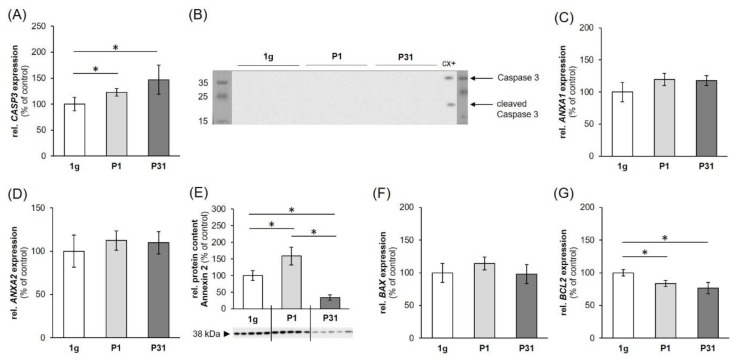
Influence of short-term microgravity on the gene expression: (**A**) *CASP3*, (**C**) *ANXA1*, (**D**) *ANXA2*, (**F**) *BAX*, (**G**) *BCL2*; and protein content (**B**) caspase 3, (**E**) annexin 2 regulators of apoptosis. The data are given as mean ± standard deviation. * *p* < 0.05 vs. 1 *g*-control. CX+ colon cancer cells served as positive control for programmed cell death.

**Figure 5 ijms-20-05730-f005:**
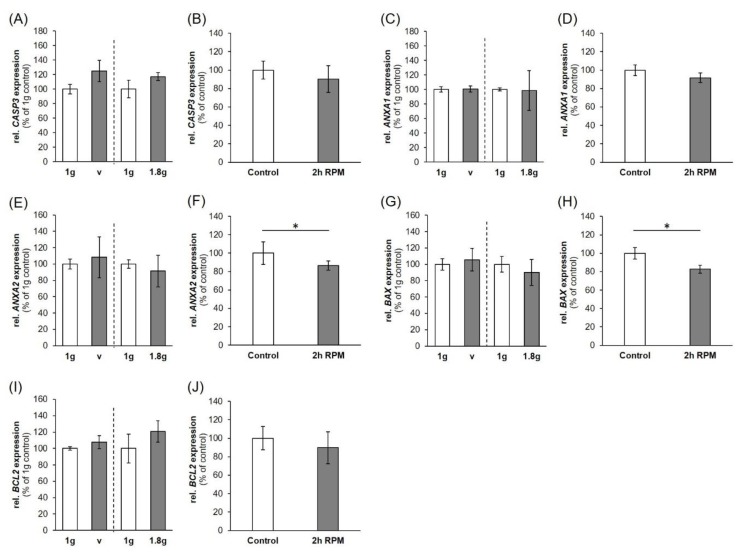
Influence of a VIB (V), hyper-*g* and iRPM-exposure on the gene expression of apoptosis signaling factors: (**A**,**B**) *CASP3*, (**C**,**D**) *ANXA1*, (**E**,**F**) *ANXA2*, (**G**,**H**) *BAX*, (**I**,**J**) *BCL2*. *n* = 5. The data are given as mean ± standard deviation. * *p* < 0.05 vs. 1 *g*-control.

**Figure 6 ijms-20-05730-f006:**
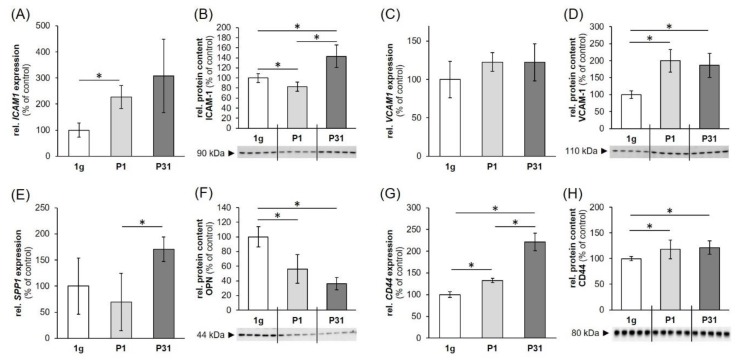
Expression of genes: (**A**) *ICAM1*, (**C**) *VCAM1*, (**E**) *SPP1*, (**G**) *CD44*; and proteins: (**B**) ICAM-1, (**D**) VCAM-1, (**F**) OPN, (**H**) CD44 involved in cell adhesion. 1 g: ground control; P1: parabola 1; P31: parabola 31. The data are given as mean ± standard deviation. * *p* < 0.05 vs. 1 *g*-control; ** *p* < 0.01 vs. 1 *g*-control.

**Figure 7 ijms-20-05730-f007:**
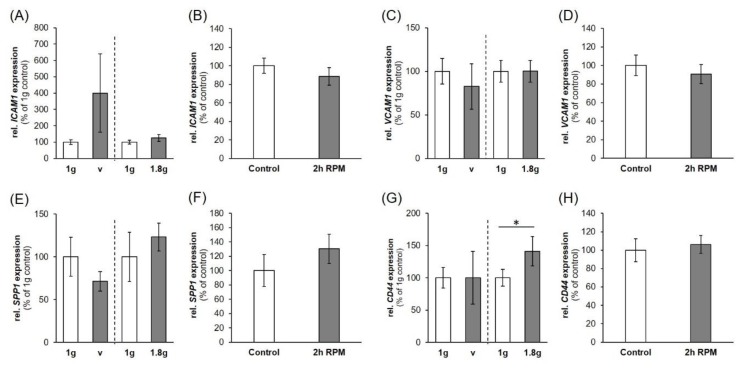
Influence of a VIB (V), hyper-*g* and iRPM-exposure on the gene expression of cell adhesion signaling factors: (**A**,**B**) *ICAM1*, (**C**,**D**) *VCAM1*, (**E**,**F**) *SPP1*, (**G**,**H**) *CD44*. 1 *g*: ground control; V: 2 h of vibration; 1.8 *g*: 2 h of hyper-*g*. The data are given as mean ± standard deviation. * *p* < 0.05 vs. 1 *g*-control.

**Figure 8 ijms-20-05730-f008:**
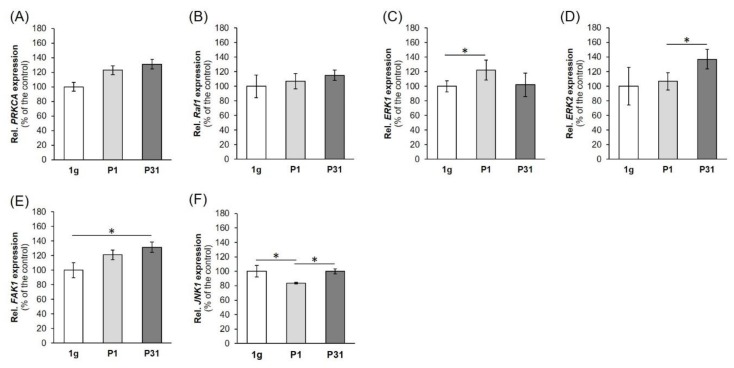
Expression of genes: (**A**) *PRKCA*, (**B**) *Raf1*, (**C**) *ERK1*, (**D**) *ERK2*, (**E**) *FAK1*, (**F**) *JNK1*, involved in cancer progression and metastasis. 1 *g*: ground control; P1: parabola 1; P31: parabola 31. The data are given as mean ± standard deviation. * *p* < 0.05 vs. 1 *g*-control.

**Figure 9 ijms-20-05730-f009:**
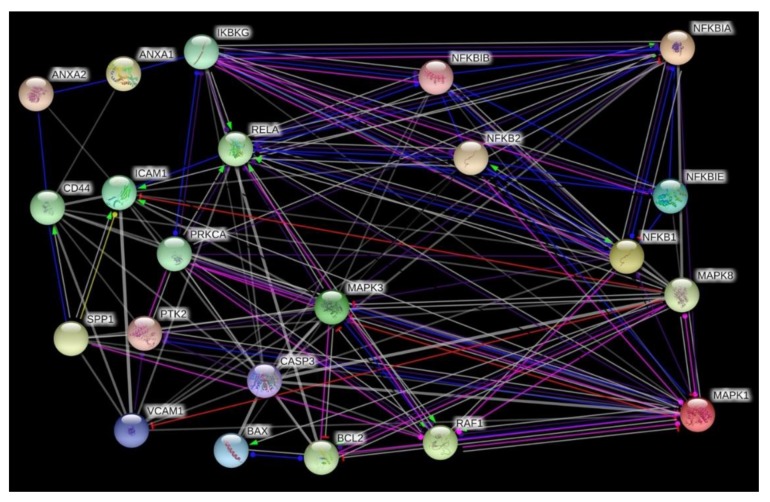
Network of the functional interaction of genes and their products analyzed in this study. The analysis was performed by STRING (Search Tool for the Retrieval of Interacting Genes/Proteins) v11.0) provided by the STRING Consortium (Available online: https://string-db.org/). The result is presented in the molecule action mode. Gene names are indicated.

**Figure 10 ijms-20-05730-f010:**
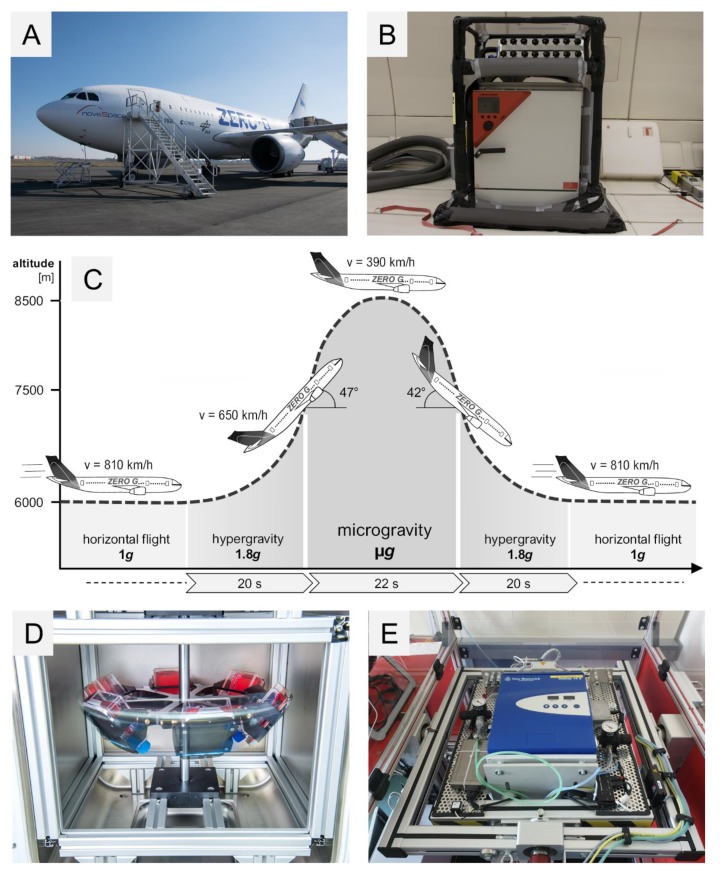
(**A**) Airbus A310 ZERO-G; (**B**) Deutsches Zentrum für Luft- und Raumfahrt (DLR) Flight rack; (**C**) Different phases of a parabolic flight; (**D**) DLR Multi-sample incubator centrifuge (MuSIC); (**E**) Incubator random positioning machine (iRPM).

**Table 1 ijms-20-05730-t001:** List of primer used during the study. Primer sequences are given in 5′ to 3′ direction.

Factor	Primer Name	Sequence 5’–3’
18S rRNA	18s-F	GGAGCCTGCGGCTTAATTT
	18s-R	CAACTAAGAACGGCCATGCA
Annexin A1; ANXA1	ANXA1-F	GCCAAAGACATAACCTCAGACACAT
	ANXA1-R	GAATCAGCCAAGTCTTCATTCACA
Annexin A2; ANXA2	ANXA2-F	GGTACAAGAGTTACAGCCCTTATGACA
	ANXA2-R	CATGGAGTCATACAGCCGATCA
BCL2 Associated X, Apoptosis Regulator; BAX	Bax-F	GTCAGCTGCCACTCGGAAA
	Bax-R	AGTAACATGGAGCTGCAGAGGAT
B-cell lymphoma 2; BCL2	Bcl2-F	TCAGAGACAGCCAGGAGAAATCA
	Bcl2-R	CCTGTGGATGACTGAGTACCTGAA
Caspase 3; CASP3	Casp3-F	CTCCAACATCGACTGTGAGAAGTT
	Casp3-R	GCGCCAGCTCCAGCAA
CD44	CD44-F	ACCCTCCCCTCATTCACCAT
	CD44-R	GTTGTACTACTAGGAGTTGCCTGGATT
Extracellular signal-regulated kinases 1; ERK1	ERK1-F	ACCTGCGACCTTAAGATTTGTGA
	ERK1-R	AGCCACATACTCCGTCAGGAA
Extracellular signal-regulated kinases 2; ERK2	ERK2-F	TTCCAACCTGCTGCTCAACA
	ERK2-R	TCTGTCAGGAACCCTGTGTGAT
Focal adhesion kinase 1 (Protein-tyrosine kinase 2); pan-FAK1	FAK1-F	TGTGGGTAAACCAGATCCTGC
	FAK1-R	CTGAAGCTTGACACCCTCGT
Intercellular adhesion molecule 1; ICAM1	ICAM1-F	CGGCTGACGTGTGCAGTAAT
	ICAM1-R	CTTCTGAGACCTCTGGCTTCGT
Mitogen-activated protein kinase 8 (MAPK8) (JNK1-a2); MAPK8/JNK1	JNK1-F	TCTCCTTTAGGTGCAGCAGTG
	JNK1-R	CAGAGGCCAAAGTCGGATCT
NF-kappa-B transcription complex P105/P50; NFKB1	NFkB1-F	CTTAGGAGGGAGAGCCCAC
	NFkB1-R	TGAAACATTTGTTCAGGCCTTC
NF-kappa-B transcription complex P100/P52; NFKB2	NFkB2-F	GTACAAAGATACGCGGACCC
	NFkB2-R	CCAGACCTGGGTTGTAGCA
NF-kappa-B transcription complex P65; NFKB P65	NFkB P65-F	CGCTTCTTCACACACTGGATTC
	NFkB P65-R	ACTGCCGGGATGGCTTCT
NF-kappa-B essential modulator (NEMO); IKBKG	IkBKG-F	AACTGGGACTTTCTCGGAGC
	IkBKG-R	GGCAAGGGCTGTCAGCAG
NF-kappa-B inhibitor alpha; NFKBIA	NFkBIa-F	AATGCTCAGGAGCCCTGTAAT
	NFkBIa-R	CTGTTGACATCAGCCCCACA
NF-kappa-B inhibitor beta; NFKBIB	NFkBIb-F	CCCGGAGGACCTGGGTT
	NFkBIb-R	GCAGTGCCGTGTCCCC
NF-kappa-B inhibitor epsilon; NFKBIE	NFkBIe-F	TGGGCATCTCATCCACTCTG
	NFkBIe-R	ACAAGGGATTCCTCAGTCAGGT
Protein kinase C alpha type; PRKCA	PRKCA-F	TGGGTCACTGCTCTATGGACTTATC
	PRKCA-R	CGCCCCCTCTTCTCAGTGT
TATA-box binding protein; TBP	TBP-F	GTGACCCAGCATCACTGTTTC
	TBP-R	GCAAACCAGAAACCCTTGCG
Raf-1 Proto-Oncogene, Serine/Threonine Kinase; Raf1	Raf1-F	GGGAGCTTGGAAGACGATCAG
	Raf1-R	ACACGGATAGTGTTGCTTGTC
Osteopontin (OPN); SPP1	SPP1-F	CGAGGTGATAGTGTGGTTTATGGA
	SPP1-R	CGTCTGTAGCATCAGGGTACTG
Vascular cell adhesion protein 1; VCAM1	VCAM1-F	CATGGAATTCGAACCCAAACA
	VCAM1-R	GGCTGACCAAGACGGTTGTATC

**Table 2 ijms-20-05730-t002:** List of the names, sources, companies, molecular weight and dilutions of all the antibodies that were used for Western blots.

Antibody	kDa	Dilution	Company	Source
Annexin 2	38	1:1000	Abcam #ab41802	Rb
NFkBp65	65	1:1000	Thermo Fisher #PA1-186	Rb
IKBKG	38	1:500	Origene #TA812460	MS
IkBα/NFKBIA	36	1:1000	Invitrogen #MA5-15132	MS
Cofilin-1	20	1:2000	Abcam #ab 42824	Rb
CD44	80	1:500	CST#5640	MS
VCAM1	110	1:500	Sc80431	MS
Casp 3	35	1:800	CST#9662	Rb
ICAM1	90	1:500	CST#4915	Rb
Osteopontin	44	1:1000	SAB4200018	MS

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
