# Peer review of "Short-Term Microgravity Influences Cell Adhesion in Human Breast Cancer Cells"

_ijms, 2019, doi:10.3390/ijms20225730_

Round 1
Reviewer 1 Report
In this manuscript, researchers have introduced an very impressive and interesting experimental study on the short-term microgravity influences on the cell adhesion in breast cancer cells. The literature searching is sufficient and supporting the technical content well. The overall writing has been well prepared. The figures can be improved a little bit by highlighting most important content with more captions. "Accept" is suggested.
Author Response
We would like to thank the reviewers for their constructive comments and hereby send you the revised manuscript.
Below you can find the reviewers’ comments and our response and the changes made to the manuscript. All changes in the manuscript are highlighted in yellow.
Reviewer I
Comments and Suggestions for Authors
In this manuscript, researchers have introduced an very impressive and interesting experimental study on the short-term microgravity influences on the cell adhesion in breast cancer cells. The literature searching is sufficient and supporting the technical content well. The overall writing has been well prepared. The figures can be improved a little bit by highlighting most important content with more captions. "Accept" is suggested.
Answer: Thank you very much for your kind review. According to your suggestion we have highlighted in red the most important results in the figures of the revised manuscript.
Reviewer 2 Report
The present manuscript reports the effect of altered gravity conditions on cell adhesion in MDA-MB-231 cells. Authors checked for the change in the expressions of genes and proteins involved in cell adhesion, apoptosis and NFkB signaling and reported upregulation of ICAM-1, VCAM-1 and CD44 and downregulation of p65 and ANAX2 levels. Overall, the paper has a potential to be accepted. However, there is a minor concern. Authors should also check any change in the expressions of upstream activators of MAPKs as MAPK signaling also contribute to metastasis. The results will strengthen the conclusion and will give a deeper understanding on the growth and function of human cancer cells exposed to altered gravity.
Author Response
We would like to thank the reviewers for their constructive comments and hereby send you the revised manuscript.
Below you can find the reviewers’ comments and our response and the changes made to the manuscript. All changes in the manuscript are highlighted in yellow.
Reviewer II
The present manuscript reports the effect of altered gravity conditions on cell adhesion in MDA-MB-231 cells. Authors checked for the change in the expressions of genes and proteins involved in cell adhesion, apoptosis and NFkB signaling and reported upregulation of ICAM-1, VCAM-1 and CD44 and downregulation of p65 and ANAX2 levels. Overall, the paper has a potential to be accepted. However, there is a minor concern. Authors should also check any change in the expressions of upstream activators of MAPKs as MAPK signaling also contribute to metastasis. The results will strengthen the conclusion and will give a deeper understanding on the growth and function of human cancer cells exposed to altered gravity.
Answer: Thank you very much for this important suggestion. We have performed additional rt PCR experiments and added a new figure 8 and have repeated the STRING analysis (see figure 9).
Data for PRKCA, RAF1, ERK1, ERK2, PTK2 and JNK1 mRNAs are included in the revised manuscript (Figure 8A-F). Please see abstract on page 1, the results on page 8, lines 198-212, page 9, as well as page 10, lines 241-244. Please see the discussion on page 14, lines 438-477 and page 15, lines 478-485.